# Recent Advances in Cell Sheet Engineering: From Fabrication to Clinical Translation

**DOI:** 10.3390/bioengineering10020211

**Published:** 2023-02-06

**Authors:** Parichut Thummarati, Wanida Laiwattanapaisal, Rikiya Nitta, Megumi Fukuda, Artchaya Hassametto, Masahiro Kino-oka

**Affiliations:** 1Department of Biotechnology, Graduate School of Engineering, Osaka University, Osaka 565-0871, Japan; 2Biosensors and Bioanalytical Technology for Cells and Innovative Testing Device Research Unit, Department of Clinical Chemistry, Faculty of Allied Health Sciences, Chulalongkorn University, Bangkok 10330, Thailand; 3Department of Pathobiology, Faculty of Science, Mahidol University, Bangkok 10400, Thailand

**Keywords:** cell sheet technique, tissue engineering, regenerative medicine, temperature-responsive polymer, transplantation

## Abstract

Cell sheet engineering, a scaffold-free tissue fabrication technique, has proven to be an important breakthrough technology in regenerative medicine. Over the past two decades, the field has developed rapidly in terms of investigating fabrication techniques and multipurpose applications in regenerative medicine and biological research. This review highlights the most important achievements in cell sheet engineering to date. We first discuss cell sheet harvesting systems, which have been introduced in temperature-responsive surfaces and other systems to overcome the limitations of conventional cell harvesting methods. In addition, we describe several techniques of cell sheet transfer for preclinical (in vitro and in vivo) and clinical trials. This review also covers cell sheet cryopreservation, which allows short- and long-term storage of cells. Subsequently, we discuss the cell sheet properties of angiogenic cytokines and vasculogenesis. Finally, we discuss updates to various applications, from biological research to clinical translation. We believe that the present review, which shows and compares fundamental technologies and recent advances in cell engineering, can potentially be helpful for new and experienced researchers to promote the further development of tissue engineering in different applications.

## 1. Introduction

Regenerative medicine aims to replace or regenerate dysfunctional cells, tissues, or organs to restore them to their original function. This is a promising research field for diseases that have, so far, been incurable. Conventional regenerative therapies, including cell suspensions, injections, scaffold-embedded cells, and other tissue engineering methods, are among the advanced approaches in regenerative medicine; however, cell loss from injection sites [1] and the lack of cell-to-cell and extracellular matrix (ECM) interactions, which significantly contribute to the properties and function of each organ and tissue, are the main challenges the techniques face [2]. This negatively affects the ability to provide signals to the cell population and promote cell adhesion, survival, and proliferation, resulting in low repair efficiency after tissue grafting [3].

Cell sheet engineering is an advanced scaffold-free tissue engineering technique applicable to repairing or regenerating defective tissues and organs, including the heart [2,4], skin [5], cornea [6], cartilage [7], esophagus [8], and brain [9]. It was first developed and published in 1990 by Yamada et al. [10]. To engineer cell sheets, the desired cells should be grown to confluence on the culture surface coated with a temperature-responsive polymer such as poly(*N*-isopropylacrylamide) (PIPAAm), which allows intact cells to be harvested without enzymatic treatment. As a result, cells are formed as monolayers along with their deposited ECM [11], intact cell surface proteins, and receptors, which play vital roles in the functional tissue. Furthermore, cell sheets can be transplanted directly into the target tissue or even used to create three-dimensional (3D) tissue-like structures [12,13]. This approach exhibits numerous advantages over conventional regenerative therapies, such as cell injection and tissue reconstruction with biodegradable scaffolds [14]. To date, cell sheet engineering has been utilized for many different applications in vitro and in vivo, as well as in clinical trials.

In addition, other new cell-based sheet fabrication techniques have been developed using multidisciplinary technologies to fabricate thicker, more functional, complex, and homogeneous tissues. This review discusses cell sheet engineering techniques, including cell sheet harvesting using temperature- and non-temperature-responsive systems, cell sheet transfer, cryopreservation, angiogenic cytokine release and vascularization, and applications for biological research and clinical uses of cell sheets.

## 2. Techniques for Harvesting Cell Sheets

One of the most important steps in generating scaffold-free tissue is harvesting intact cells from the culture surface. Platforms should be designed to allow cells to adhere and detach without digesting the ECM, which acts as a glue between cell layers. Additionally, the platform should preserve all signaling proteins and molecules important for promoting cellular functions and biological processes. Various systems have been designed to harvest cell sheets without treatment with proteases such as trypsin. To date, several technologies have been explored to harvest cell sheets. These include temperature-responsive systems using synthetic polymers and non-temperature-responsive systems, such as the ion-induced cell-detachment method, electro-responsive systems, photo-responsive systems, pH-responsive systems, mechanical systems, and magnetic systems. In this section, we describe the principles by which each platform promotes intact cell recovery from the culture surface and its corresponding efficiency. Examples of cell sheet fabrication methods using different platforms are presented in Table 1.

### 2.1. Temperature-Responsive Systems Using PIPAAm

Several synthetic polymers with molecular architectures are responsive to changes in pH, electric field, chemical species, and temperature, among other environmental factors. PIPAAm is a material of great interest and is widely used in biomedical fields owing to its unique behavior: reversible solubility in an aqueous solution with a change in temperature, as first reported by Heskins and Guillet [26]. Polymer chains of PIPAAm hydrate to form an expanded structure in water at temperatures below its lower critical solution temperature (LCST) of 32 °C. However, these chains form a compact structure upon dehydration at temperatures higher than the LCST. This transition of the aqueous PIPAAm solution is mainly due to conformational changes in the polymer chain arising from hydration changes in the isopropyl side groups [27].

To harvest cell sheets, Okano’s research group pioneered the development of temperature-responsive culture plates by grafting a PIPAAm layer on a tissue culture polystyrene surface (TCPS) through the electron beam polymerization method [10,28] (Figure 1A). At 37 °C, the culture surface is hydrophobic, and cells are allowed to attach and proliferate. By decreasing the temperature to 20 °C, the culture surface becomes hydrophilic, and the cells easily detach from the surface [10,29,30] (Figure 1B). However, detachment of cell sheets from surfaces of TCPS grafted with PIPAAm is slow (~75 min to detach primary bovine aortic endothelial cells (BAECs)), occurring gradually from the periphery of the sheet toward the interior. Thus, a longer time is required to detach an intact cell sheet completely. In principle, the limiting step in detaching cell sheets is the diffusion of water molecules beneath the PIPAAm-grafted surface. Therefore, approaches have been considered to accelerate the hydration of hydrophobic PIPAAm chains. These include the introduction of a highly water-permeable microporous membrane between the cell sheet and the PIPAAm surface, reducing the detachment time of BAECs to within 30 min [15]. Other approaches to accelerate cell sheet detachment include grafting PIPAAm with poly(ethylene glycol) (PEG) onto porous cell culture membranes (from which BAECs take 19 min to detach) [17], developing comb-type grafted PIPAAm gels on a TCPS (taking 25 min for BAECs to detach) [18] and grafting poly(2-hydroxyethyl methacrylate) (PHEMA) and PIPAAm onto TCPS (from which BAECs take 30 min to detach) [19]. Furthermore, modification of the PIPAAm culture surface for rapid cell sheet harvesting has been demonstrated by grafting an ultrathin double polymeric nanolayer consisting of PIPAAm and hydrophilic polyacrylamide (PAAm) (30 min for BAEC detachment) [20]. The latter design has the benefit of modulating the surface properties of specific cells, which could be useful for harvesting various types of cells. Copolymerization of IAAm with various types of hydrophilic or hydrophobic monomers (such as methacrylate (BMA) and PEG) is another approach to modify the temperature-responsive surface to control cell attachment/detachment [31,32] and cell patterning [33]. Mitsuhiro E. et al. demonstrated that grafting Poly(IAAm-co-carboxyisopropylacrylamide (CIPAAm)) on TCPS accelerated detachment of BAECs, reducing the detachment time to 35 min, compared to a control PIPAAm dish (from which BAECs took 60 min to detach) [21]. To mimic specific tissue functions, it is important to fabricate cell sheets from multiple cell types using a co-culture system. However, the proliferation and adhesion properties varied among cell types because of the differences in expression levels and types of adhesion molecules on the cell surface [34]. Therefore, the micropatterned temperature-responsive surface was designed to harvest cell sheets containing multiple cell types. An example of IAAm copolymer for cell patterning was demonstrated by Tsuda Y. et al. In this system, poly(IPAAm-co-BMA) on TCPS dishes was grafted on TCPS dished using the electron beam irradiation method, and cell attachment/detachment could be modulated by varying the BMA content [35].

PIPAAm-grafted surfaces have shown to be non-toxic and biocompatible when tested in vitro with various types of cells, including endothelial cells, epithelial cells, smooth muscle cells, and fibroblasts [36,37]. Harvesting of co-culture cell sheets has been successfully demonstrated using a PIPAAm-grafted system [38,39]. Numerous studies have successfully demonstrated the fabrication of cell sheets from various types of cells using a temperature-responsive PIPAAm-based polymer for preclinical and clinical trials. This system provides a highly reproducible and stable surface for temperature-responsive cell cultures.

However, grafting a PIPAAm-based polymer onto culture surfaces is complicated and requires special equipment, which is technically difficult and very costly. TCPS culture surfaces grafted with PIPAAm are already commercially available, making the preparation of cell sheets with a few steps, but these culture surfaces are much more expensive than general cell-culture vessels.

### 2.2. Temperature-Responsive System Using Methylcellulose (MC)

MC is a water-soluble polysaccharide derived from cellulose by partially substituting hydrophilic hydroxyl groups with methoxy groups. The phase transition between the MC–water solution and the gels is characterized by the LCST, which varies depending on the concentration of MC in the aqueous solution [40] and the addition of salts [41]. This property is associated with a change in hydrophilicity at temperatures below the LCST, hydrogen bonding between water and MC hydroxyl groups, hydrophobicity at temperatures above the LCST, dehydration via the exposure of methoxy groups, and stronger interactions among them. Additionally, MC is easy to use, inexpensive, and biocompatible. These features make MC a promising material for cell sheet fabrication and tissue engineering.

In addition, MC-based temperature-responsive surfaces have been designed to overcome the challenges of the complicated and time-consuming process of electron beam irradiation for PIPAAm grafting. In 2006, Chen et al. demonstrated a simple and inexpensive method to fabricate a cell sheet using an 8% *w*/*v* MC solution mixed with distinct salts (e.g., Na_2_SO_4_ and phosphate-buffered saline) on TCPS dishes at room temperature (~20 °C), which subsequently gelled at 37 °C (MC hydrogel) (Figure 2A). The gel at 37 °C was coated with neutral aqueous collagen at 4 °C for cell attachment. Cells were allowed to grow confluently on the MC/PBS/collagen surface to form a monolayer. The cell sheets were harvested at 20 °C without proteolytic enzyme treatment (Figure 2B). Cell sheets (containing human foreskin fibroblasts) were completely removed after 20 and 10 min by shaking [22]. However, the procedure described was complex, did not produce uniform hydrogels, and was consistently unstable because the MC formulation used was too viscous to be easily manipulated. To address this challenge, the optimal composition of the MC/PBS/collagen surface was systematically investigated. The production of stable hydrogels was dependent on the molecular weight (MW) and concentration of MC, as well as the type and concentration of the added salt. The optimal combination of MC–water–salt was found to be 12–16% of MC (MW = 15,000 g/mol) in 0.15 M PBS (~150 mOsm). Following this procedure, an MC gel was formed at ~32 °C. Detachment of the entire cell sheet was completed at room temperature (30 °C) for 2–3 min. Furthermore, monolayer and thick multilayer cell sheets were successfully constructed [23].

Similar to PIPAAm, the MC surface also provides a non-toxicity platform for various cell types, including stem cells [42]. However, a comparative study demonstrated that MC may decrease cell proliferation in cell sheets after culture for more than 2 weeks, while cells on the PIPAAm surface can continue to proliferate [42]. The advantages and disadvantages of responsive systems are listed in Table 2.

### 2.3. Non-Temperature Responsive Systems Using Ion-Induced Cell Detachment

The ion-induced cell detachment method was designed as a simple isothermal system to detach cells at the desired time. This system does not require an electron beam or vapor-phase polymerization equipment or facilities to graft the cell-culture surface [49]. Since a temperature-responsive surface can damage highly sensitive cells, alter cellular metabolism, and change the cell cycle, gene expression, and function [50], an ion-induced cell detachment surface is safer for cultured cells. The principle of this system is based on the fact that cell-substrate adhesion can be regulated by modulating the surface energy of the cell-culture substrate: high surface energy promotes cell adhesion, and vice versa [51,52,53]. Therefore, this approach is composed of two platforms to generate a non-responsive cell sheet, modulate cell–substrate adhesion, and trigger the detachment of cells at the designed time without cell damage. To control cell–substrate interactions, a copolymer film comprising nonpolar hydrophobic divinylbenzene (DVB) and polar hydrophilic 4-vinylpyridine (4VP) was generated on a tissue culture surface using the initiated chemical vapor deposition (iCVD) method [54]. The surface of the copolymer film can be adjusted by controlling the input flow ratio from DVB to 4VP. These surface properties can be used to modulate the adsorption of ECM proteins, such as fibronectin, on surface-modified culture plates (Figure 3A).

The trigger method to release cell sheets from the culture surface involves the conformational change of integrin, which is a family of cell surface proteins that modulates cell–substrate or cell–ECM interactions, upon the depletion of divalent cations such as Ca^2+^ and Mg^2+^ [55]. Therefore, cell sheets can be harvested under isothermal conditions at the desired time by adding Dulbecco’s phosphate-buffered saline (DPBS) without Ca^2+^ and Mg^2+^ (Figure 3B). The detachment of the cell sheet could be accelerated to within 100 s. In this study, there were five types of cells, including thermosensitive cell types (myoblast cell line (C2C12) and fibroblast cell line (NIH-3T3 cells)) and non-thermosensitive cell types (mesenchymal stem cells (MSCs)), normal human dermal fibroblasts (NHDFs) and endothelial cell line (C166)), used to evaluate the efficiency of the developed method. These cells were completely detached as intact cell sheets. In addition, this non-responsive system has been used to fabricate monolayer and thick multilayer cell sheets in both in-vitro and in-vivo studies. Moreover, the previous study confirmed that the pDV surface did not significantly affect cell–cell adhesion and cell viability [24].

### 2.4. Non-Temperature-Responsive Systems Using Electro-Responsive Surfaces

Another common approach for recovering cell sheets is to utilize an electro-responsive surface. In principle, electrical stimulation is the signal that triggers cell detachment in this system. The major component of this system is a self-assembled monolayer (SAM) of alkanethiolates on gold connected to immobilizing peptide ligands containing Arg-Gly-Asp (RGD), a binding site for cell adhesion. The types of immobilizing ligands can be designed to adapt to various cells for adhesion. When a negative electrical potential is applied to the gold film, the monomer is oxidized, resulting in the rapid release of the immobilized ligands [45]. The detachment of the cell sheets from these surfaces was completed by applying −1.0 V, and cells became completely detached within 10 min, faster than on temperature-responsive surfaces. Electro-responsive systems are adaptable and are comprised of different cell-culture surfaces, allowing the creation of varying cell sheet sizes with different forms and thicknesses [56].

Enomoto et al. demonstrated the use of an electro-responsive surface to prepare fibroblast sheets with a thickness of ~50 μm. The oligopeptide CCRRGDWLC, which contains RGD, was designed for ligand immobilization. Fibroblasts grew rapidly on the membrane for 14 days, and the thickness of the cell sheet became ~60 μm, which is almost three times thicker than that of cells cultured in conventional cell-culture plates. Subsequently, the stacking of these cell sheets to up to five layers established multiple thick cell sheets with a total thickness of more than 200 μm [57]. However, a necrotic core began to develop in these cell sheets. This problem can be overcome by generating a continuous flow of the culture medium around the stacked sheets to provide better oxygenation and nutrient provision [57].

Another experiment to create thick tissues was conducted by Kobayashi et al. They combined an electric-responsive platform with microstereolithography to create a thick three-dimensional tissue with a complicated shape. The idea behind these types of cell sheets is to apply them to repair the intestinal wall, as they can be made to fit their anatomical features in a precise manner. The advantage of these 3D cell sheets is their use for the repair of more complex structures and the regeneration of more complex organs [25].

### 2.5. Other Systems

A photo-responsive system requires a fabrication method which uses light as a trigger to control wettability. Light can illuminate and reversibly change the conformation of photo-responsive materials due to changes in various properties, including magnetism, fluorescence, and wettability. Among these properties, light has attracted attention as a cell-adhesion controller. Metal oxides, primarily zinc oxide (ZnO) and titanium dioxide (TiO_2_), are the most widely used photo-responsive materials. Hong et al. demonstrated the use of a photo-responsive cell sheet system. They designed a cell sheet system by coating a TiO_2_ nanodot-coated quartz substrate onto a cell-culture plate, and pre-osteoblastic cells were seeded until confluency. After UV-light (365 nm) illumination, the cell sheets were completely detached within 20 min [46]. This evidence demonstrates that a photo-responsive system is a promising method for harvesting cell sheets.

Furthermore, pH-responsive systems have been widely used in drug delivery systems because of the pH variability in the human body, which can control the release of drugs to the target area. Classic examples of this system include cancer-drug delivery systems. pH-responsive systems allow anti-cancer drugs to be released at the tumor site, where the pH is approximately 6.5–7.2, while the pH of the physical condition is 7.4. However, few studies have used pH-responsive systems for cell sheet fabrication due to the limited range of pH values (6.8–7.4) allowed for normal cell function. Chen et al. developed a pH-responsive chitosan surface to control cell detachment within a small pH range. HeLa cells attached to the surface of chitosan at pH 6.99 and 7.20. After the pH was increased to 7.65, almost all cells detached from the surface within 1 h and survived [48].

Various responsive systems have been developed to enable the detachment of confluent cell sheets; however, the limitations of these systems require further study. These include the evaluation of the potential of these systems with numerous cell types and their extension for various applications. Some cells that attach firmly to a surface may have a longer detachment time. In addition, commercially responsive surfaces are already available for cell sheet detachment; however, they are costly, especially when thick 3D tissue constructs need to be fabricated using a large number of cell sheets. Another challenge is that most current fabrication technologies remain inaccessible and complicated. Therefore, the development of simple and economical fabrication methods for responsive surfaces is necessary and will greatly encourage researchers to exploit cell sheet engineering.

## 3. Cell Sheet Transferring Techniques

Cell sheet engineering has been applied in various in-vitro and in-vivo experiments and clinical trials. To date, various techniques to transfer cell sheets have been developed and should be considered for each application. These techniques include pipetting, hydrogel plungers, and membranes. The details of each technique and its limitations are discussed in this section.

### 3.1. Simple Pipetting Method

Pipetting is the simplest method to transfer cell sheets to the transplant area or to create thick multi-layered 3D tissue. Cell sheets should detach and shrink upon transferring because of their intercellular tension. Most sheets became smaller in diameter but thicker after detachment. Then, sheets containing entire cells together with the culture media are carefully sucked inside a serological pipette or Pasteur pipette and transferred to another culture dish. Cell sheets may be folded on a new surface after aspiration from the pipette. To spread cell sheets, a complete culture medium can be added dropwise to the transferred cell sheets. After spreading the cell sheets, the culture medium should be removed, and the cell sheets should be incubated at 37 °C for 30 min to allow adhesion (Figure 4A). To stack the next layer, these steps are repeated. Centrifugation of the culture plate allows the cell sheet to attach more rapidly to a new surface [58]. Although this technique is very simple, inexpensive, and requires special equipment, manipulating cell sheets requires skill, as cell sheets are fragile during manipulation.

### 3.2. Gelatin Plungers or Stamps

Gelatin is the most common hydrogel used to prepare cell sheet stamps because it can form a gel at low temperatures, at which cell sheets are detached from temperature-responsive surfaces. Furthermore, this hydrogel is non-toxic and can be easily removed from the stacked cell sheet via incubation at 37 °C, which is the temperature used to culture cells. In addition to the pipetting method, gelatin plungers or stamps can be placed on a cell sheet before detachment from the culture surface. The protocol for preparing a gelatin plunger was first developed by Okano et al. Briefly, a 7.5% *w*/*v* gelatin solution was prepared in Hank’s balanced salt solution (HBSS) and incubated in a water bath at 55 °C to dissolve the gelatin. The final volume of the solution was adjusted to 10 mL and the gelatin solution was neutralized with 200 μL of 1 N NaOH. For sterilization, the gelatin solution was filtered using a 0.45 μm membrane filter [59]. To create a gelatin plunger, the gelatin solution was coated with a manipulator or loaded inside a silicone rubber mold [60]. The latter was required immediately after sealing the gelatin solution inside the silicone mold rubber. Gelatin was allowed to solidify at a cool temperature by placing the gelatin plunger or stamp inside a sterilized plastic box and placed in a refrigerator at 4 °C overnight to form a gel [13]. To stack the cell sheets, gelatin plungers or stamps were overlaid onto the cell sheets prior to detachment. The gelatin plunger or stamp was then lifted with the cell sheet from the bottom surface of the wells and transferred to the next layer or a new surface (Figure 4B). These steps were repeated when preparing multi-layered cell sheets. This method is easy to manipulate during the transfer of cell sheets, as cell sheets can be prevented from being fragile; however, many gelatin preparation steps are required. This method was suitable for cell sheet fabrication and transplantation in in-vitro studies.

### 3.3. Membrane-Assisted Transfer

The principle of membrane-assisted transfer is based on the adhesion of the cell sheet to the membrane via capillary pressure, whereas the adhesion to another cell sheet or target tissue is due to the presence of adhesion molecules [61]. This approach is easy to perform, and no residues of external materials remain at the transfer site. Therefore, this technique is widely used for in-vivo transplantation. To harvest the cell sheet and transfer it using this technique, membranes are overlaid on the cell sheets before they are allowed to detach, similar to gelatin stamping. The culture medium is removed during cell detachment from the responsive surface. After detachment, the cell sheets and the membranes are transferred to a new surface or transplantation site. The membranes are, finally, removed after the cell sheets are allowed to adhere to the transplanted area (Figure 4C). A drop of cell-culture medium over the membrane promotes membrane release from the cell sheet. Examples of membranes used for cell sheet transfer that are widely available in almost all molecular laboratories are polyvinylidene difluoride (PVDF) and nitrocellulose membranes [24,62]. Although PVDF and nitrocellulose membranes are rigid and difficult to place properly on curved sites, they are still used for transplantations on the cornea [63], the middle ear [64], and wounds [24]. CellShifter™ is commercially available and one of the most-cited membranes used for in-vitro and in-vivo cell sheet transfer [65,66,67]. Another commercial membrane used for cell sheet transplantation is Seprafilm^®^ (sodium hyaluronate and carboxymethylcellulose) [68]. This membrane is available as a surgery material and can be left in a surgical site because it is biocompatible and resorbs within 7 days.

## 4. Cryopreservation

Cryopreservation is a critical process in the field of regenerative medicine. To date, various types of cell sheets have demonstrated safety, feasibility, and efficacy in clinical practice [69,70,71,72,73]. However, limitations may arise when the required point of use is separated by distance and, more frequently, by time from the facilities where the cells were isolated and prepared. To address these limitations, designing the conditions for short-term and long-term preservation is necessary not only for single cells before fabrication but also for the final cell sheet product. Although there is currently a 3D tissue cryopreservation protocol that includes cell sheets, it remains challenging to preserve the structure and function of cell sheets after storage and to develop simple methodologies for routine clinical applications [74,75]. In this review, we describe two major methodologies for cell sheet cryopreservation: slow-freezing and vitrification.

Currently, the most common slow-freezing protocol involves keeping cells in the presence of dimethyl sulfoxide (DMSO). The freezing solution is mainly composed of DMSO and serum and should be cooled before the cells are mixed. The freezing tubes are then kept in a freezer at low temperatures (between −20 °C and −80 °C) or in a nitrogen tank (−196 °C). For high cell viability, cells should be slowly frozen at 0.5–1 °C/min [76,77] or using a device such as a Program Freezer [78]. In addition, slow-freezing has been used for cell sheet cryopreservation [79,80,81]; however, slow freezing at −20 °C and −80 °C takes up to several hours but still provides only short-term preservation. For long-term storage, cell sheets should be transferred to liquid nitrogen. The main advantage of slow freezing is its simplicity and availability and the lower risk of sample contamination because all procedures can be performed in a cabinet using sterile reagents.

Vitrification is an alternative methodology to cryopreserve cell sheets without the formation of ice crystals, which can damage the cells. Compared to the slow-freezing method, a much higher concentration of cryopreservative agents, a much more rapid freezing rate at a very low temperature (at −80 °C or −196 °C in liquid nitrogen), and the sequential adding of cryopreservative agents are required to make the solution inside and surrounding the cells become viscous for vitrification. This method was first developed by Rall and Fahy in the 1980s to preserve the structure and functions of mammalian organs for long-term storage [82]. Nowadays, several types of cryopreservative agents with different components and osmolarity have been developed for the storage of cell sheets, as shown in Table 3.

However, most vitrification methods are complex work processes and risk contamination because they include an opened system. Therefore, the conventional vitrification method might be inappropriate for clinical-application uses. Therefore, improvement in the vitrification method is required to develop a simpler and more practical method for clinical application. Recently, Hayashi et al. developed ‘a circulating vitrification bag’ made from a sealable polyethylene which allows the sequential flush of solutions of the vitrification process in a closed system. In addition, a vitrification storage box was developed for long-term preservation of up to 6 months for transportation of the rabbit chondrocyte sheet. The box was a stainless-steel container with built-in LN absorbents in the main body and the lid [83].

**Table 3 bioengineering-10-00211-t003:** Components of vitrification agents for cell sheet storage.

Types of the Cell Sheet	Vitrification Solution Component	Storage Period	Year, Refs
Chondrocyte cell sheet	20% dimethyl sulfoxide, 20% ethylene glycol (EG), 0.5 M sucrose, and 10% carboxylated poly-L-lysine	3–6 months	2013, [84] 2020, [83]
Mesenchymal stem-cell sheet	6.5 M EG, 0.5 M sucrose, and 10% *w*/*w* carboxylated poly-l-lysine (COOH-PLL) in PBS	-	2016, [85]
Skeletal-muscle myoblast cell sheet	6.5 M EG, 0.7 M sucrose, and 10% carboxyl poly-L-lysine	3 months	2018, [86]
Oral-mucosa epithelial cell sheets	2.5%, 5%, and 10% of EG	204 days	2019, [87]
	Vitrification solution 1 (1.7% *w*/*v* EG, 1.3% *w*/*v* formamide, 2.2% *w*/*v* DMSO, 0.7% *w*/*v* PVP K12, and 0.1% *w*/*v* of SuperCool X-1000 and SuperCool Z-1000, Vitrification solution 2 (4.7% *w*/*v* EG, 3.6% *w*/*v* formamide, 6.2% *w*/*v* DMSO, 1.9% *w*/*v* PVP K12, and 0.3% *w*/*v* of SuperCool X-1000 and SuperCool Z-1000) and Vitrification solution 3 (16.84% *w*/*v* EG, 12.86% *w*/*v* formamide, 22.3% *w*/*v* DMSO, 7% *w*/*v* PVP K12, and 1% *w*/*v* of SuperCool X-1000 and SuperCool Z-1000)	204 days	2019, [87]

## 5. Cytokines and Vascularization in Cell Sheets

Vasculogenesis and angiogenesis are important processes that promote new blood vessel formation. Vasculogenesis refers to the in-situ differentiation and growth of blood vessels from precursor cells into endothelial cells and the formation of a primitive vascular network, which refers to the new formation of blood vessels from the existing site [88]. These processes are essential for animal development and the healing of damaged tissues [89].

At the single-cell level, the growth of endothelial cells from existing blood vessels is controlled by balancing various positive and negative stimuli, which drive internal signaling responses and their resultant effects, including destabilization, invasion, migration, proliferation, cell-to-cell connection, branching, and capillary formation [90,91]. The main stimuli for these processes are cytokines, including various growth factors such as vascular epithelial growth factor (VEGF), fibroblast growth factor (FGF), hepatocyte growth factor (HGF), and platelet-derived endothelial cell growth factor (PD-ECGF) [92,93,94,95].

In terms of tissue engineering, the angiogenic potential is one of the main goals when designing a tissue engineering platform. The survival of cells in the middle of a thick three-dimensional tissue construct depends on the vascular supply of oxygen and nutrients [96]. Furthermore, the network has the potential to connect the vasculature of the patient, resulting in accelerated perfusion of implant-engineered tissues [97]. Consequently, the induction of angiogenic factors to promote the formation of blood vessels in engineered tissue constructs is a major challenge for the broad clinical use of tissue engineering applications. Several approaches are currently under investigation to improve vascular growth in implanted tissue constructs [98,99]. Various approaches have also been studied to promote angiogenic potential in engineered tissues, including cell sheets, to accelerate vasculogenesis. Approaches to promote rapid vasculogenesis in tissue constructs can be classified into increased positive angiogenic stimulus production and prevascularization.

Gene modification and co-culture systems have been shown to increase the production of angiogenic stimuli. Augustin et al. overexpressed VEGF in an MSC sheet using the gene modification method. VEGF-modified MSC sheet therapy was then transplanted into the rat ischemic myocardium, and a functional myocardium was recovered [100]. Using a similar technique, myoblast sheets were then transfected with VEGF plasmids using poly(β-amino ester) nanoparticles. This study demonstrated increased angiogenic potential in an animal model, providing a more effective therapeutic result than the control after myocardial ischemia [101]. This also confirms that angiogenesis can be enhanced by promoting the production of angiogenic cytokines in cell sheets.

Cell sheets can be co-cultured by randomly mixing or stacking layers of different cell types. Co-culture of different cell types not only enhances cytokine secretion but also modulates cell behavior and function in tissues [38]. Kino-oka M. and colleagues demonstrated a system to promote VEGF and HGF secretion in monolayer and multilayer cell sheets by co-culturing fibroblasts with human-skeletal-muscle myoblasts. The results demonstrated that monocultures of skeletal muscle, fibroblasts, and myoblasts produced low levels of VEGF, but only fibroblasts produced high levels of HGF. Upon co-culture of these two cells at a suitable proportion within the cell sheet, VEGF and HGF secretion was enhanced, significantly increasing angiogenesis in vitro [38]. Okano T. and colleagues proposed the use of double-layer co-culture cell sheets. In this method, endothelial cell sheets were directly overlaid onto hepatocyte cell sheets. After several days of culture, hepatocytes in the cell sheet were maintained in a differentiated cell shape and expressed albumin [39]. This evidence confirmed that a co-culture system could maintain cytokine production, cell behavior, and function.

Although skin and cartilage tissue products are commercially available, most are limited to thin tissues. The engineering of larger and more complex tissues could potentially improve therapeutic efficiency for the treatment of critical diseases [102]. To date, several pre-vascularization strategies have been employed to create functional engineered tissues. Another pre-vascularization system was proposed by Okano T. and colleagues. In this system, endothelial cells are sandwiched between myoblast sheets. In-vitro studies demonstrated that endothelial cells inside the sheet elongated and formed a capillary-like structure [59]. In addition, pre-vascularized multilayer cell sheets were transplanted into an animal model. The graft and host tissues were successfully connected, indicating that successful transplantation was achieved [60]. These studies strongly confirmed that cell sheets could undergo pre-vascularization, promoting tissue function and vasculogenesis.

## 6. Applications

To date, cell sheet technology has been utilized for a wide range of applications. In this section, we describe the applications of cell sheet technology from in-vitro experiments to in-vivo and clinical studies. In addition, the use of cell sheet technology to support various biological research is also described.

### 6.1. Clinical Translation

Since cell sheet engineering has been successfully developed, cell sheets from various types of cells have been manufactured for various tissues and the treatment of various diseases, such as periodontitis, ocular trauma, skin burning, osteoarthritis, middle-ear disease, and myocardial infarction (MI) (Table 4). Among these, MI has been the most studied over the last two decades. In this section, we summarize the achievements in cell sheet engineering for clinical applications, mainly focusing on cell sheet engineering for MI.

MI is one of the leading causes of death worldwide [103]. Undoubtedly, heart tissue cannot regenerate itself upon damage due to its lack of cell-dividing ability. Recently, the use of regenerative medicine and tissue engineering has gained importance, not only as a new method of treating severe heart failure but also as an effective approach to overcome the poor prognosis of patients with heart failure.

**Table 4 bioengineering-10-00211-t004:** Cell sheet fabrication and clinical applications.

Cell Types	Fabrication Methods	Clinical Treatment	Stage of Study	Year, Refs.
Skeletal-muscle cells	PIPAAm surface	Ischemic myocardium	Preclinical in vitro	2006, [104]
				2013, [105,106]
				2020, [38]
			Preclinical in vivo	2005, [107] 2014, [108]
			Clinical study	2012, [73]
				2015, [4]
				2017, [70]
				2021, [109]
Myoblasts	PIPAAm surface	Pancreatic fistula	Preclinical in vivo	2017, [110]
Fibroblasts	PIPAAm surface	Ischemic myocardium	Preclinical in vivo	2008, [111]
		Wound ulcer		
Mesenchymal stem cells	PIPAAm surface	Ischemic myocardium	Preclinical in vivo	2012, [100]
				2014, [112]
				2016, [113]
	Ion-induced surface	Ischemic limb	Preclinical in vivo	2020, [24]
	MC surface	Bone regeneration	Preclinical in vitro	2022, [114]
	Thermo-responsive EMDs	Ocular trauma	Preclinical in vitro	2022, [115]
	PIPAAm surface	Diabetic ulcers	Preclinical in vivo	2015, [116]
Embryonic stem cells	PIPAAm surface	Ischemic myocardium	Preclinical in vivo	2012, [117]
iPS cells	PIPAAm surface	Ischemic myocardium	Preclinical in vitro	2012, [118]
			Preclinical in vivo	2012, [119]
				2014, [120]
				2018, [121]
	PIPAAm surface	Liver failure	Preclinical in vivo	2016, [122]
	PIPAAm surface	Stroke and brain damage	Preclinical in vivo	2017, [123]
Osteogenic cells	PIPAAm surface	Bone regeneration	Preclinical in vitro	2017, [42]
	MC surface	Bone regeneration	Preclinical in vitro	2017, [42]
Corneal epithelium	PIPAAm surface	Ocular trauma	Preclinical in vitro	2004, [124]
Oral mucosal epithelial cells	PIPAAm surface	Esophageal ulcer	Preclinical in vitro	2010, [125]
		Esophageal ulcer	Preclinical in vivo	2021, [126]
		Esophageal neoplasm	Clinical study	2012, [127]
		Ocular trauma	Clinical study	2004, [6]
Keratinocytes	PIPAAm surface	Skin defect	Preclinical in vivo	2017, [128]
				2019, [129]
Keratinocytes fibroblasts	PIPAAm surface	Burn wound	Preclinical in vitro	2020, [130]
Periodontal-ligament (PDL)-derived cells	PIPAAm surface	Periodontitis	Preclinical in vitro and in vivo Clinical study	2010, [131] 2018, [132]

Over the past decade, several studies have demonstrated the safety, feasibility, and efficacy of cell sheet engineering for the treatment of several diseases, including MI. To date, many types of cells have been used for the cell sheet-based transplantation of hearts damaged by MI or cardiomyopathy, including skeletal myoblasts [133,134,135,136], MSCs [137], adipocytes [138,139], dermal fibroblasts [140,141], and iPS [142]. Among these, skeletal myoblasts are the most interesting cells for engineering heart tissue because of the abundance of muscles and their ability to allow autologous transplantation. Okano’s group at Tokyo Women’s Medical University, Japan, pioneered the establishment of autologous skeletal-muscle myoblasts using a temperature-responsive system [107]. Skeletal-muscle cell sheets were fabricated, and several experiments were performed to confirm the efficacy, feasibility, and functional recovery of the heart in vitro, in vivo, and clinical trials [143].

Regarding the treatment mechanism, myoblast sheets could produce several cytokines, such as VEGF, HGF, and fibroblast growth factor 2 (FGF-2), which are essential factors for actively promoting angiogenesis in the infarcted area and attracting progenitors to improve the damaged part, resulting in the induction of cardiac tissue regeneration [144]. Multilayer myoblast sheets, such as three- and five-layer sheets, promoted favorable results with better angiogenesis induction, the presence of more elastic fibers, and less fibrosis compared to monolayer sheets, probably due to increased cytokine secretion from the HSMM [145]. From these findings, a thicker skeletal muscle cell sheet should be applied clinically because more cytokine secretion in the infarcted area could induce angiogenesis and recover the damaged heart. In 2017, autologous skeletal stem cell sheets of ~100 μm in thickness and ~4 cm in diameter were fabricated from skeletal-muscle cells and transplanted into patients with severe heart failure. This phase-I clinical trial demonstrated the safety, feasibility, and functional recovery of the heart after transplantation [71].

Most studies on tissue engineering have used MSCs. These cells exist in almost all tissues, including the bone marrow, adipose tissue, and synovium [146]. They are easily extracted from tissues and have the potential to differentiate into various target cells. Similar to skeletal-muscle myoblasts, MSCs are ideal for autologous transplantation [147] and several studies have demonstrated the potential of MSCs in regenerative medicine. In addition to MSCs, induced-pluripotent-stem-cell (iPSC)-derived cardiovascular cell populations are novel cell sources for cardiac regeneration. Recently, hiPSC-derived cardiac-tissue sheets were manufactured, showing therapeutic efficacy in vitro and in vivo [121]. The results showed that hiPSC-derived cardiac-tissue sheets provide positive outcomes for cardiac regenerative therapy. This evidence demonstrated that cell sheet engineering may provide a possible solution for the clinical repair of various damaged organs, including the impaired myocardium (Figure 5).

### 6.2. Models for Biological Research

Cancer is a leading cause of death worldwide. It has been reported that almost 1 in 4 people died in the United States due to cancer in 2005 [148]. In-vitro models for cancer have been used to increase our knowledge of cancer biology and help develop drugs for the pharmaceutical industry. Numerous novel anti-cancer drugs have been discovered over the past half-century. Unfortunately, more than 80% of the anti-cancer compounds that appear promising in two-dimensional in-vitro models have failed in various phases of clinical trials due to the unrealistic conditions of these models compared to the actual target tissues, which comprise heterogeneous cell populations and anatomical complexity [149,150,151]. Animal models have been used to evaluate drug efficacy and toxicity, but they often fail to translate to clinical cancer trials; they typically have a success rate of less than 8% due to biological differences between humans and animals [152]. To address this issue, the integration of new advanced techniques in tissue engineering has begun to yield three-dimensional tissue models that more accurately mimic the native tumor microenvironment and are physiologically translatable, constituting a paradigm shift in drug discovery.

Cell sheet technology is suitable for the construction of dense and thick tissues. Furthermore, temperature-responsive plates (such as UpCell) are commercially available; therefore, cell sheets can be fabricated in any biological laboratory. Much evidence has been obtained using cell sheet engineering as an anti-cancer drug screening model.

Conventional in-vivo cancer models have been developed through the injection of cancer cells into animals. However, this approach is not always successful. In 2014, Suzuki et al. developed a cell sheet-based cancer model using four different cell lines to evaluate anti-cancer therapeutics, including colon (HCT-116), pancreas (Panc-1), liver (Li-7), and stomach cells (MKN74) [153]. Subsequently, these cell sheets were transplanted into nude mice and compared with the conventional cell-suspension injection method in terms of their ability to form tumors. Cancer cell sheets provided more stable engraftment and showed a larger tumor volume. In addition, a similar study was performed by Alshareeda et al. using sheets of hepatocellular carcinoma and stromal cells [154]. The Yang research group reported the tissue thickness-dependent cytotoxic efficacy of the anti-cancer drug doxorubicin using cell sheet-based liver models [155]. Furthermore, a co-culture of keratinocyte cell sheets (keratinocytes) and cancer spheroids (cancer + cancer-associated fibroblasts) has been designed to mimic the tumor microenvironment to test several chemotherapeutic drugs for resistance [156].

Angiogenesis is associated with tumor progression and is particularly important in cancer modeling. Angiogenesis stimulates tumor tissues to gain more nutrients and oxygen [157]. Li et al. developed a 3D in-vitro angiogenesis model using cell sheet technology. In this model, multi-layered cell sheets were constructed from myoblasts and co-cultured with seeded human umbilical vein endothelial cells (HUVECs). After 24 h, HUVECs moved from the bottom of the culture surface to form a capillary-like network. The structure was visualized in both 2D and 3D. In addition, in this study, a quantitative method was developed to analyze the capillary-like network [105]. This model was considered a promising in-vitro angiogenesis model not only for evaluating transplant efficiency but also for cancer angiogenesis. To confirm this model, a multi-layered cell sheet prepared by mixing a small population of RMS cells with skeletal-muscle myoblasts was constructed to demonstrate the ability of cancer cells to invade adjacent muscles and angiogenesis [158]. Taken together, this evidence confirms that cell sheet engineering is a powerful tool for drug discovery and cancer biology (Figure 5).

## 7. Conclusions and Future Direction

Cell sheet engineering is an advanced tissue engineering technique in regenerative medicine. In this technology, enzyme treatment is not required for cell harvesting; therefore, ECM and cell adhesion molecules are well-preserved. The key elements of cell sheet technology include cell sheet harvesting, transferring, cryopreservation, and prevascularization. The details of these elements were discussed in this review. Various systems, including temperature-responsive and non-temperature-responsive systems for cell sheet harvesting were described, focusing on the complexity, biocompatibility, and detachment efficiency.

Cell sheet transferring is another crucial step in cell sheet engineering. We described the details of current techniques, including simple pipetting, gelatin plunger or stamp, and membrane-assisted transfer methods, as well as the limitations of each method. This section will be useful for considering how to choose the suitable transferring method for individual applications.

Cryopreservation is another important aspect of regenerative medicine, as it allows the preservation of engineered tissues for future use. Currently, the most common method of cell sheet cryopreservation is slow-freezing, which involves keeping cells in the presence of DMSO and cooling them at low temperatures. This method is simple and available but provides only short-term preservation. An alternative method is a vitrification, which uses a high concentration of cryopreservative agents and a rapid freezing rate to prevent the formation of ice crystals. However, this method is complex and risky for contamination, and it requires improvement to be more practical for clinical applications. Recently, a circulating vitrification bag and storage box have been developed to improve the vitrification method for the long-term preservation and transportation of cell sheets. Although various studies on cell sheet techniques have been elucidated, cryopreservation of cell sheets requires further investigation to obtain high-quality cell sheets after long-term storage and transportation.

Vasculogenesis and angiogenesis are important processes which promote the formation of new blood vessels, which are essential for the healing of damaged tissues. Cytokines, including growth factors such as VEGF, FGF, HGF, and PD-ECGF, play a crucial role in endothelial cell growth and the formation of a primitive vascular network. In tissue engineering, the method to induce the angiogenic potential in the cell sheet remains a major challenge. Various approaches have been studied to improve the secretion of growth factors in the cell sheet, including gene modification and co-culture systems. A co-culture system of different cell types not only enhances cytokine secretion but also modulates cell behavior and function in tissues. However, more research needs to be carried out to develop a more practical and efficient method for promoting angiogenesis and vascularization in the cell sheet after transplantation to the host tissue.

As cell sheets preserve cell-to-cell and ECM interactions which resemble microenvironments for cells and are essential for cell functions, cell sheet engineering is a promising cell-based therapy and can be applied in a wide range of regenerative medicines and biomedical modeling. These include treatment for MI, many epithelial defect diseases, and cancer models for drug testing. Recently, it was reported that cell sheet technology can be applied to the food industry to produce meat from isolated muscle cells [159]. However, only a few studies have been performed; additional studies are required for further investigation.

One of the most-studied applications of cell sheet technology is for the treatment of tissue infarction, including MI, which is one of the leading causes of death worldwide. Cell sheets from various types of cells have been manufactured for the treatment of MI, including skeletal myoblasts, MSCs, adipocytes, dermal fibroblasts, and iPSCs. Studies have demonstrated the safety, feasibility, and efficacy of cell sheet engineering for the treatment of MI, with the use of autologous skeletal muscle myoblasts being a promising approach. Cell sheets can produce several cytokines, such as VEGF, HGF, and FGF-2, which play a crucial role in promoting angiogenesis in the infarcted area and attracting progenitors to improve the damaged part, resulting in the induction of cardiac-tissue regeneration. However, more research needs to be carried out to develop a more practical and efficient method for cell sheet-based transplantation in clinical settings.

Although various cell sheet engineering techniques, including cell sheet harvesting, cell sheet transfer, cryopreservation, and vascularization, have been developed to fabricate more functional and complex cell sheets for transplantation and representation of disease models, these techniques still have limitations. For example, the variation in detachment time depends on the type of cells, how to effectively prevascularize more complex tissues and a short-/long-term cell sheet cryopreservation method that can preserve cell sheet structure and function. Therefore, further research that covers such limitations will be beneficial for future applications of cell sheet engineering in both regenerative medicine and biomedical research.

## Figures and Tables

**Figure 1 bioengineering-10-00211-f001:**
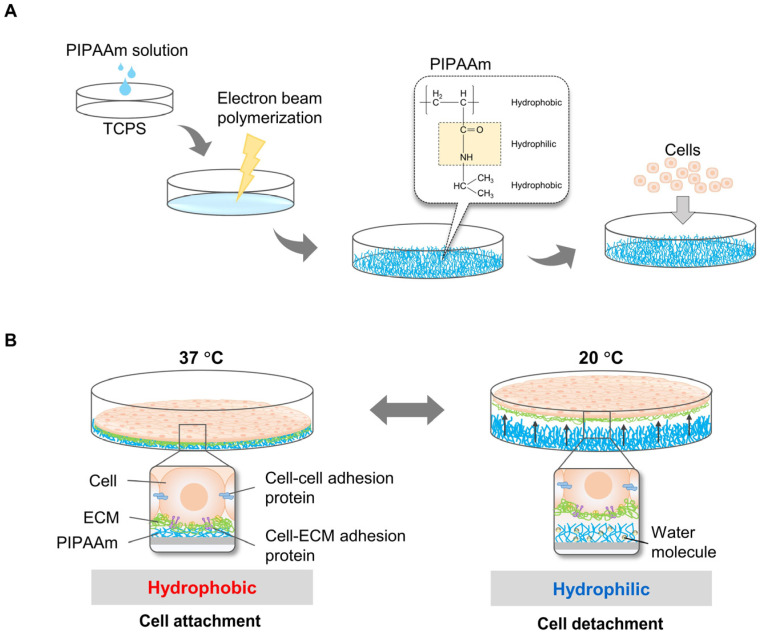
Principles of cell sheet harvesting using PIPAAm grafted surface. PIPAAm was grafted on the TCPS surface by electron beam polymerization before cells were allowed to grow on the surface. (**A**) Cells attached and proliferated on the PIPAAm cultured surface incubated at 37 °C, higher than the LCST of 32 °C. To harvest the cell sheet, cells were incubated at 32 °C below the LCST. PIPAAm was hydrophilic at this temperature (**B**).

**Figure 2 bioengineering-10-00211-f002:**
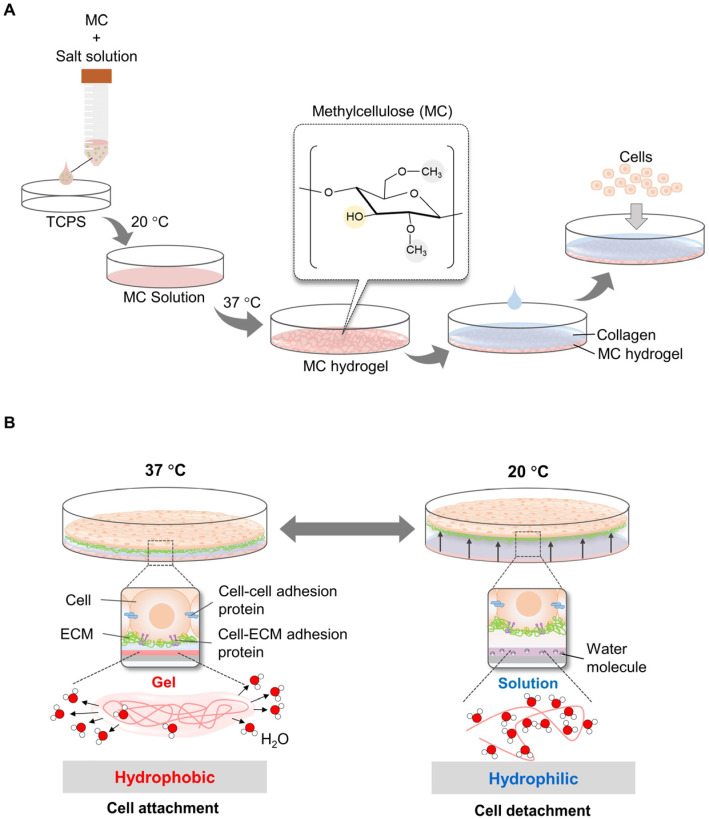
Principles of cell sheet harvesting using MC/collagen-coated surface. MC mixed with salt solution (Na_2_SO_4_ or PBS) was poured on the TCPS surface and gelled at 37 °C. Subsequently, neutral collagen at 4 °C was coated on the MC surface to increase cell attachment efficiency (**A**). Cells on MC/collagen-coated surface were grown at 37 °C, which MC was hydrophobic. By decreasing the temperature to 20 °C, MC was transformed and bonded with water molecules. As a result, cells were detached from the MC surface (**B**).

**Figure 3 bioengineering-10-00211-f003:**
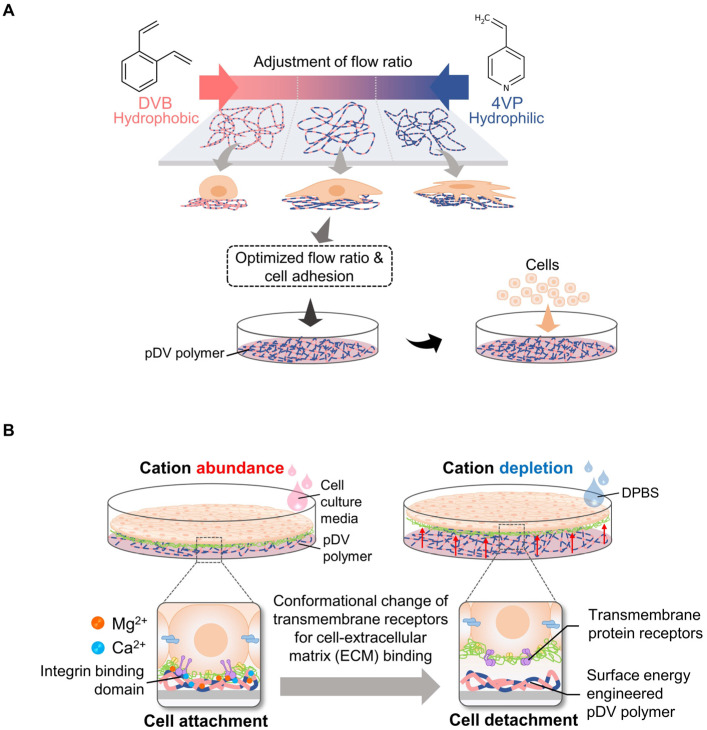
Principles of cell sheet harvesting using engineered pDV film coupled with divalent cation depletion. The flow ratio of DVB and 4VP was optimized for cell attachment efficiency and copolymerized on a cultured surface (**A**). To trigger cell detachment, the cell environment was depleted in bivalent cations (Mg^2+^ and Ca^2+^) by pouring DPBS into cell culture dishes or plates. Consequently, transmembrane proteins responsible for cell ECM occurring were conformationally changed and unbound from the binding domain. As a result, cells were detached from the culture surface (**B**).

**Figure 4 bioengineering-10-00211-f004:**
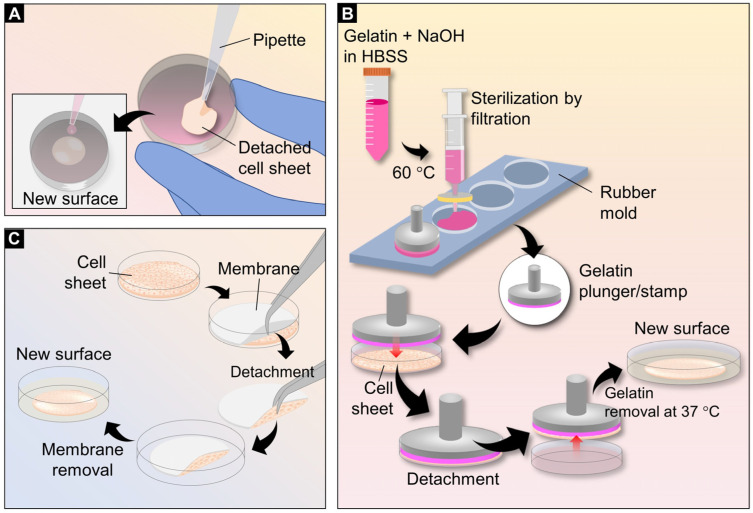
Different techniques for cell sheet transfer. The simple technique was introduced by sucking detached cell sheets inside a serological pipette or Pasteur pipette. The cell sheets were subsequently transferred to the new surface. The folded cell sheets were spread by dropwise addition of culture media (**A**). To prepare plungers or stamps, gelatin powder in Hank’s balanced salt solution mixed with 1N NaOH was incubated at 60 °C to dissolve the gelatin completely. The gelatin solution was sterilized using a membrane filter with 0.45 μm in pore size. Subsequently, the gelatin solution was added to the rubber mold. Plungers or stamps were inserted into the gelatin solution in the mold. The gelatin was gelled at 4 °C overnight. The gelatin plungers or stamps were placed directly on the cell sheet. After transferring, gelatin was dissolved at 37 °C and removed from the cell sheet (**B**). Membranes such as PVDF and nitrocellulose membranes were used to transfer cell sheets. Similar to the gelatin stamp technique, the membranes were placed directly onto the cell sheets. However, this technique does not require the gelatin removal step and is most suitable for in-vivo and clinical studies (**C**).

**Figure 5 bioengineering-10-00211-f005:**
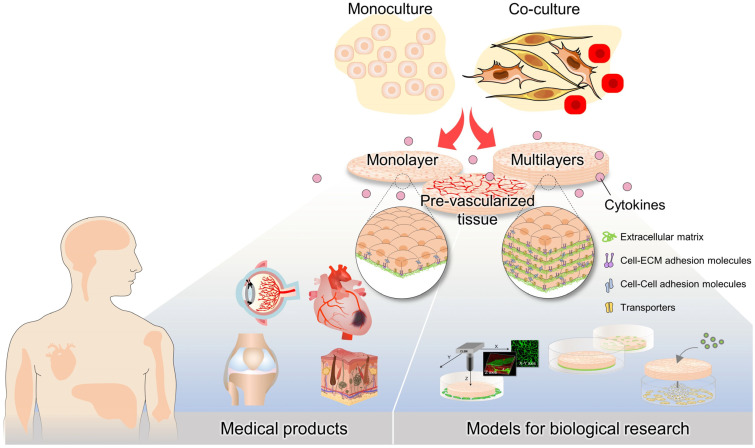
Applications of cell sheet engineering. Monolayer and multilayered cell sheets were fabricated from monocultures or cocultured systems. To increase the grafting rate after transplantation, cell sheets were constructed with prevascularization by coculture with endothelial cells. Recently, cell sheet engineering has been studied not only for medical products but also models for biological research.

**Table 1 bioengineering-10-00211-t001:** Cell sheet harvesting system and detachment time.

Cell Types	Responsive System on TCPS	Detachment Temperature/Time	Refs.
BAECs	PIPAAm on TCPS	20 °C/~75 min	[15,16]
	PIPAAm/microporous membrane	20 °C/~30 min	[15]
	PIPAAm/PEG/microporous membrane	20 °C/~19 min	[17]
	A comb-type grafted PIPAAm	20 °C/~25 min	[18]
	PIPAAm/PHEMA	20 °C/~30 min	[19]
	PIPAAm/PAAm Poly(IAAm-co-CIPAAm)	20 °C/~30 min 20 °C/~35 min	[20] [21]
Dermal fibroblast	MC/PBS/Col (8% MC, PBS MW = 77,000–94,000, 10 g/L PBS)	20 °C/~10–20 min	[22]
Human-adipose-tissue-derived stem cells	MC/PBS/Col (12% to 16% MC, MW = 15,000, 1.5 M PBS)	RT (~30 °C)/~2–3 min	[23]
Dermal fibroblast, MSC, myoblasts, endothelial cells	DVB/4VP/Ion-induction	37 °C is possible/~100 s	[24]
Dermal fibroblasts	Electrical responsive system	37 °C is possible/~5 min	[25]

**Table 2 bioengineering-10-00211-t002:** The advantages and disadvantages of the different types of platforms for cell sheet engineering.

Responsive Systems	Advantages	Disadvantages	Refs
PIPAAm-grafted surface	It is commercially available. Cell detachment is highly effective. Multilayer cell sheets can be prepared. Biocompatibility and feasibility have been tested in vivo and clinical trials. PIPAAm grafting system is adaptable to obtain more complex tissue structures and functions. Applications in vivo and clinical trials have been demonstrated. Detachment of the co-culture cell sheet has been demonstrated. Patterned grafting onto TCPS dishes has been demonstrated.	The grafting method is complicated and time-consuming. Special equipment (such as an electron beam) is required for grafting. Highly costly method. Temperature change may affect the cell cycle and metabolism. Commercial dishes are much more expensive than general dishes. Detachment time varied depending on cell types.	[10,28,36,38,39,42,43,44]
MC-coating surface	The coating method is simple and inexpensive. Cell detachment is highly effective. The culture surface is reusable. The coating method does not require special equipment.	It is not commercially available. Coated MC may disintegrate and swell. MC may decrease cell proliferation. Optimization of coating-solution composition may be required for each cell type. Temperature change may affect the cell cycle and metabolism. Detachment time varied depending on cell types.	[22,42]
Ion-induced cell detachment	Isothermal system. It is suitable for highly sensitive cells. The culture surface is reusable. Inexpensive system.	Exposure of cells to ion depletion buffer may affect cell metabolisms and signaling pathways. Detachment time varied depending on cell types.	[24]
Electro-responsive surface	High precision in cell patterning. Inexpensive system.	Electrochemical dissolution of polyelectrolyte coatings can cause local pH changes, which may be harmful to sensitive cells. Properly immobilized ligands need to be designed for different cell types.	[25,45]
Photo-responsive surface	Inexpensive system.	pH range is limited to 6.8–7.4 for the normal function of the cell. The potential of the system needs to be evaluated with various cell types.	[46]
pH-responsive system	Inexpensive system.	Detachment may be incomplete in some types of cells. The potential of the system needs to be evaluated with various cell types. Few studies have been demonstrated. Culture cells can cause the pH to change.	[47,48]

## Data Availability

Not applicable.

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
