# Peer review of "Recent Advances in Cell Sheet Engineering: From Fabrication to Clinical Translation"

_bioengineering, 2023, doi:10.3390/bioengineering10020211_

Round 1
Reviewer 1 Report
The review "Recent Advances in Cell Sheet Engineering: from Fabrication to Clinical Translation" demonstrates a new outlook on the problems in cell sheet engineering. The review is well-structured and includes a useful informational summary. Without a doubt, the review should be published in Bioengineering, but only after significant modification. Certain points need to be explained.
The outlook in the future is absent in the review. This is a crucial section of the review since it helps us predict how the subject will evolve in the future.
The advantages and disadvantages of the different types of platforms for cell sheet engineering should be compared.
Can the same cell sheet engineering platforms be used to successfully cultivate different types of cells? Appropriate discussion should be added to the review.
Various IPAAm-based copolymers for cell sheet engineering have been developed in recent years. Copolymers-based platforms have some advantages in comparison to PIPAAm-based systems. This information should be added to the review.
The authors left out information about temperature-responsive systems based on oligo(ethylene glycol) methacrylates for cell sheet engineering. (https://doi.org/10.1002/adma.201100597; https://doi.org/10.3390/polym14194245).
It is necessary to provide information about the potential toxicity of the systems presented here.
Finally, please cite some relevant papers that may improve the quality of the review:
https://doi.org/10.3390/polym14194245
https://doi.org/10.1002/adma.201100597
https://doi.org/10.1016/j.porgcoat.2022.107376
https://doi.org/10.3390/ma14061417
Author Response
Dear Reviewer,
We sincerely thank the Reviewer to review of our manuscript and for the excellent suggestions that we received. We have made a concerted effort to adequately respond to each suggestion received from the Reviewer. We firmly believe that the Reviewer’s comments and suggestions have significantly improved this manuscript. We do hope that you and the Reviewer find this manuscript acceptable for publication in Engineering.
- The outlook in the future is absent in the review. This is a crucial section of the review since it helps us predict how the subject will evolve in the future.
Answer: Thank you very much to point this out. We agree with this comment. Therefore, we describe this content in “section 7: Conclusion and future direction” line no. 596-661.
- The advantages and disadvantages of the different types of platforms for cell sheet engineering should be compared.
Answer: Thank you for this suggestion. We added this formation in Table 2 and the context in line no. 136-140 and 172-176.
- Can the same cell sheet engineering platforms be used to successfully cultivate different types of cells? Appropriate discussion should be added to the review.
Answer: Thank you very much. We agree with this suggestion. We have this information in Table 3 already and added the discussion in the text line no. 119-123, 172-173, and 222-227.
- Various IPAAm-based copolymers for cell sheet engineering have been developed in recent years. Copolymers-based platforms have some advantages in comparison to PIPAAm-based systems. This information should be added to the review.
Answer: Thank you very much for pointing this out. We described this information in line no. 113-127.
- The authors left out information about temperature-responsive systems based on oligo (ethylene glycol) methacrylates for cell sheet engineering. (https://doi.org/10.1002/adma.201100597; https://doi.org/10.3390/polym14194245)
Answer: Thank you very much for pointing this out and suggesting references. This information was described in line no. 104-106 and 114.
- It is necessary to provide information about the potential toxicity of the systems presented here.
- Finally, please cite some relevant papers that may improve the quality of the review:
https://doi.org/10.3390/polym14194245
https://doi.org/10.1002/adma.201100597
https://doi.org/10.1016/j.porgcoat.2022.107376
https://doi.org/10.3390/ma14061417
Answer: Thank you very much for the suggestion. We added:
- https://doi.org/10.3390/polym14194245 in Ref no. 45
- https://doi.org/10.1002/adma.201100597 in Ref no. 30
Sincerely,
Masahiro Kino-oka
Department of Biotechnology, Graduate School of Engineering, Osaka University,
2-1 Yamadaoka, Suita, Osaka 565-0871, Japan
Email address: kino-oka@bio.eng.osaka-u.ac.jp
Reviewer 2 Report
In this review paper, the author summarizes the progress and achievements of cell sheet engineering to date.
Major:
This review paper is well written, but the conclusion is missing, giving the impression that the article is incomplete. It would be nice to add the overall conclusion at the end.
Minor:
There are some typos in this manuscript. Please correct and revise them carefully.
Page3 line 115: temperature responsive -> temperature-responsive
Page5 line 160: detached from the MC surface (B). . -> detached from the MC surface (B). (double period)
Page7 line 234: the biding domain -> the binding domain
Page10 line 333: clinical studies. . -> clinical studies. (double period)
Author Response
Dear Reviewer,
We sincerely thank the Reviewer to review of our manuscript and for the excellent suggestions that we received. We have made a concerted effort to adequately respond to each suggestion received from the Reviewer. We firmly believe that the Reviewer’s comments and suggestions have significantly improved this manuscript. We do hope that you and the Reviewer find this manuscript acceptable for publication in Engineering.
Major:
- This review paper is well written, but the conclusion is missing, giving the impression that the article is incomplete. It would be nice to add the overall conclusion at the end.
Answer: Thank you very much to point this out. We agree with this comment. Therefore, we describe this content in “section 7: Conclusion and future direction” line no. 596-661.
Minor:
- There are some typos in this manuscript. Please correct and revise them carefully.
Answer: Thank you very much for pointing this out. We have checked the typos and revised them following your suggestion
- Page3 line 115: temperature responsive -> temperature-responsive
Answer: Thank you very much for pointing this out. We have revised it following your suggestion.
- Page5 line 160: detached from the MC surface (B). . -> detached from the MC surface (B). (double period)
Answer: We are sorry for the mistakes. We deleted “.” following your suggestion. Thank you very much.
- Page7 line 234: the biding domain -> the binding domain
- Answer: We are sorry for the mistakes. We changed from “the biding domain to the binding domain” following your suggestion. Thank you very much.
- Page10 line 333: clinical studies. . -> clinical studies. (double period)
Answer: We are sorry for the mistakes. We deleted “.” following your suggestion. Thank you very much.
Sincerely,
Masahiro Kino-oka
Department of Biotechnology, Graduate School of Engineering, Osaka University,
2-1 Yamadaoka, Suita, Osaka 565-0871, Japan
Email address: kino-oka@bio.eng.osaka-u.ac.jp
Round 2
Reviewer 1 Report
The authors have answered all of my comments and the paper can be accepted in its present form.
Reviewer 2 Report
The authors have revised all review points, this paper suitable for publication.